EMBO
*reports*

# CD1d-mediated activation of group 3 innate lymphoid cells drives IL-22 production

Julia Saez de Guinoa[1,†], Rebeca Jimeno[1,†], Nazanin Farhadi[1], Peter J Jervis[2], Liam R Cox[3], Gurdyal S Besra[2] & Patricia Barral[1,*]

## Abstract

**Innate lymphoid cells (ILCs) are a heterogeneous family of immune cells that play a critical role in a variety of immune processes including host defence against infection, wound healing and tissue repair. Whether these cells are involved in lipid-dependent immunity remains unexplored. Here we show that murine ILCs from a variety of tissues express the lipid-presenting molecule CD1d, with group 3 ILCs (ILC3s) showing the highest level of expression. Within the ILC3 family, natural cytotoxicity triggering receptor (NCR)⁻CCR6⁺ cells displayed the highest levels of CD1d. Expression of CD1d on ILCs is functionally relevant as ILC3s can acquire lipids *in vitro* and *in vivo* and load lipids on CD1d to mediate presentation to the T-cell receptor of invariant natural killer T (iNKT) cells. Conversely, engagement of CD1d *in vitro* and administration of lipid antigen *in vivo* induce ILC3 activation and production of IL-22. Taken together, our data expose a previously unappreciated role for ILCs in CD1d-mediated immunity, which can modulate tissue homeostasis and inflammatory responses.**

**Keywords** CD1d; IL-22; ILC; NKT cell
**Subject Category** Immunology

## Introduction

Innate lymphoid cells (ILCs) are a family of immune cells that play a central role in a variety of physiological processes including immune regulation, modulation of interactions with the microbiota, tissue repair and lymphoid tissue development [1]. ILCs can be classified in three families based on their cytokine secretion and the transcription factors required for their development: ILC1s (T-bet⁺) produce IFN-γ; ILC2s (GATA-3⁺) secrete IL-5 and IL-13; and ILC3s (RORγt⁺) produce IL-17 and IL-22 [2]. Accordingly, ILCs can modulate immune cell function through cytokine secretion but also through direct cell–cell interactions. Recent studies have identified

an unexpected function for ILCs as antigen presenting cells (APCs) in the regulation of CD4⁺ T-cell immunity. As such, both ILC2s and ILC3s can internalize and present antigens on MHC-II to control cognate T-cell activation [3–7], T helper 17 cell differentiation [8] and the intestinal selection of commensal bacteria-specific CD4⁺ T cells [9]. Concurrently, ILC–T-cell crosstalk promotes ILC activation and cytokine production [4,5], evidencing a bidirectional dialogue between the innate and adaptive immune systems.

Similar to the MHC-dependent recognition of protein antigens by conventional T cells, a broad range of lipids associated with CD1d molecules specifically activate a population of unconventional T cells called invariant natural killer T (iNKT) cells [10,11]. Consequently, iNKT cells recognize through their T-cell receptors (TCR) CD1d-restricted self-lipids as well as lipids from pathogenic bacteria, commensals or fungi, contributing to the establishment of immune homeostasis and to anti-microbial, anti-tumour and autoimmune responses [10,11]. Evidence suggests that the context of lipid presentation provided by diverse CD1d⁺ APCs can shape the outcome of immune responses, by tuning the extent of iNKT cell activation, their proliferation and their pattern of cytokine secretion [12–14]. Conversely, iNKT cell–APC interactions can modulate APC activation and function [14–20]. For instance, iNKT cell activation in response to glycolipid recognition results in CD40 ligand-dependent maturation of dendritic cells (DC), favouring cross-presentation and enhancing CD4, CD8 and B-cell responses [18–20]. Remarkably, engagement of CD1d is sufficient to induce activation and secretion of cytokines by both professional (DCs, monocytes) and unconventional (intestinal epithelial cells; IECs) APCs resulting in modulation of immunity in homeostasis and inflammation [14,21–23]. For example, ligation of CD1d in IEC leads to secretion of IL-10 rendering protective effects in murine models of inflammatory bowel disease (IBD) [14,21].

ILCs are emerging as central regulators of immunity, yet their possible contribution to lipid-dependent immune responses has never been explored. Here we show that ILCs from murine tissues express the lipid-presenting molecule CD1d with the highest levels of expression corresponding to ILC3s. Using fluorescent lipids and presentation assays, we demonstrate that primary ILC3s are able to internalize and load lipids on CD1d for recognition by the TCR of

1 The Peter Gorer Department of Immunobiology, King's College London, London, UK
2 School of Biosciences, University of Birmingham, Edgbaston, Birmingham, UK
3 School of Chemistry, University of Birmingham, Edgbaston, Birmingham, UK
*Corresponding author. Tel: +44 2071883060; E-mail: patricia.barral@kcl.ac.uk
†These authors contributed equally to this work

iNKT cells. Conversely, engagement of CD1d on ILC3s *in vitro* and administration of antigenic lipids *in vivo* induce ILC3 activation and secretion of IL-22. Thus, our data reveal a previously unknown function for ILC3s on CD1d-dependent immunity.

# Results and Discussion

## ILCs express CD1d

To investigate the expression of the lipid-presenting molecule CD1d by ILCs, we first identified them as lineage (Lin)$^-$CD45$^+$CD127$^+$ cells and then used the expression of T-bet, GATA-3 and RORγt to discriminate the ILC1, ILC2 and ILC3 populations, respectively (Figs 1 and EV1). We detected CD1d expression in the putative ILC population (Lin$^-$CD45$^+$CD127$^+$) within the mesenteric lymph nodes (mLN), spleen, Peyer's patches (PP), small intestinal lamina propria (SI-LP), colonic LP and lung from WT mice in comparison with ILCs from CD1d-deficient mice (Fig EV1A). The level of CD1d expression in ILCs was generally comparable to DCs and at least 10 times higher than CD45$^-$ cells (Fig 1A). In all analysed tissues, RORγt$^+$ ILCs exhibited the highest levels of CD1d amongst the ILC populations, while T-bet$^+$ ILCs showed the lowest levels (Fig 1B). Previous studies have shown that intestinal RORγt$^+$ ILCs display a gradient of T-bet that controls their fate and function and ILC3s can up-regulate T-bet and down-regulate RORγt acquiring functional and phenotypical features of ILC1s [24,25]. Accordingly, we detected that intestinal RORγt$^+$ ILCs show variable expression of T-bet, which correlated inversely with the levels of CD1d (Figs 1B and EV1B and C). Finally, analysis of gene expression by quantitative PCR showed CD1d mRNA expression on ILC3s sorted from the tissues of WT mice in comparison with ILC3s isolated from CD1d-deficient mice (Fig 1C). Thus, ILCs express CD1d with the higher levels of expression corresponding to RORγt$^+$ ILC3s.

The ILC3 population is heterogeneous and comprises several subfamilies with distinctive phenotypic and functional characteristics including: (i) lymphoid tissue inducer (LTi) cells (CCR6$^+$NKp46$^-$) which are crucial for the formation of secondary lymphoid organs and produce lymphotoxin, IL-17A and IL-22. LTi cells include CD4$^-$ and CD4$^+$ cells; (ii) natural cytotoxicity triggering receptor (NCR)$^-$ ILC3s (NKp46$^-$) secrete IL-17A and/or IL-22 and have been associated with inflammatory conditions. NCR$^-$ ILC3s can give rise to NCR$^+$ ILC3s in a process dependent on T-bet; and (iii) NCR$^+$ ILC3s (NKp46$^+$) that secrete IL-22 but not IL-17A and participate in the regulation of intestinal homeostasis [2,26]. To interrogate the levels of CD1d expression in the different ILC3 subpopulations, we performed additional phenotypical analyses of ILC3s from different tissues (Figs 1D and E, and EV1–EV3). As previously described, we found that the majority of RORγt$^+$ ILC3s in the mLN were NCR$^-$CCR6$^+$ cells [7] which expressed high levels of CD1d (Fig 1D and E) and lymphotoxin-alpha (*LTA*) mRNA but lacked T-bet (Fig EV1C and D). Within the SI-LP, we found low CD1d expression in NCR$^+$ and NCR$^-$CCR6$^-$ ILC3s in comparison with NCR$^-$CCR6$^+$ cells (Fig 1D and E). Finally, CD1d$^{hi}$ ILC3 showed high expression of CD117 (c-kit), lacked PLZF, and showed heterogeneous expression of MHC-II, CD4, and CD90, which were also differentially expressed on RORγt$^+$ ILCs from various tissues

(Figs EV2 and EV3). Thus, ILCs express CD1d with the higher levels of expression corresponding to NCR$^-$CCR6$^+$ ILC3s.

RORγt$^+$CCR6$^+$ ILC3s are a developmentally independent subset originally described to promote lymphoid organogenesis and which are major producers of IL-22 in homeostatic conditions and during infection [24,27]. In the intestine, RORγt$^+$CCR6$^+$ ILC3s expressing MHC-II control intestinal homeostasis through cognate interactions with CD4$^+$ T cells [3,8,9]. In the adult spleen, RORγt$^+$CCR6$^+$ ILC3s promote CD4$^+$ T-cell responses and enhance antibody production by marginal zone B cells [4,28]. The fact that CD1d expression is particularly high in RORγt$^+$CCR6$^+$ ILC3s points towards a possible contribution of ILC3s to CD1d-mediated immunity either in the establishment of mucosal homeostasis or during inflammatory or infectious immune responses.

## ILC3s internalize lipids and mediate CD1d-dependent lipid presentation

Next, we investigated the capacity of ILCs to internalize and load lipids on CD1d for presentation to the TCR of iNKT cells (Fig 2). Because ILC3s express the highest levels of surface CD1d (Fig 1), we focussed on this population for our functional assays.

First, we tested the ability of ILC3s to acquire lipids by incubating sort-purified ILC3s with a fluorescent synthetic form of the CD1d-binding iNKT cell-activating model lipid αGalCer (BODIPY-αGalCer; Fig 2A). After incubation with fluorescent lipids *in vitro*, ILC3s exhibited an increase in fluorescence intensity (comparable to control DCs) indicative of an ability to acquire lipid antigens, while lipid uptake was severely impaired when incubation was performed at 4°C (Fig 2A). Since ILC3s are located in the marginal zone of the spleen [28] where lipids can be efficiently retained when arriving in the circulation [29], we tested whether splenic ILC3s have the capacity to acquire lipid antigens *in vivo* (Fig 2B). To investigate this, we injected fluorescent lipids intravenously in WT mice and analysed lipid acquisition by ILC3s and DCs by flow cytometry. We observed that 16 h after lipid injection, a proportion of splenic ILC3s and DCs showed an increase in fluorescence intensity revealing their ability to acquire lipids *in vivo* (Fig 2B). Multiple mechanisms may collaborate to mediate lipid internalization by ILC3s and such processes often are determined by the nature of the antigen (i.e. lipoproteins, particulate material, pathogens). A variety of receptors expressed by many cell types (such as low-density lipoprotein receptor (LDL-R) or scavenger receptors) have been proposed to mediate lipid antigen uptake and direct such lipids towards the CD1d presentation pathway [30–32]. For instance, exogenous lipids can be incorporated into VLDL particles, which can then be internalized through LDL receptor-mediated uptake [31,32]. Moreover, the uptake of pathogens or particulate material by phagocytosis delivers exogenous lipid antigens into the endocytic system, where CD1d molecules can bind them [30]. Importantly ILC3s can internalize proteins and latex beads [3,4] as well as lipids, which suggests that they may be able to use a variety of mechanisms to sample their environment.

Next, we examined the capacity of ILC3s to load lipids on CD1d and to present them to iNKT cells (Fig 2C–F). Sort-purified ILC3s (from mLN or spleen) were pre-incubated with lipids for 2 h and co-cultured with iNKT hybridoma cells (DN32.D3). As shown in Fig 2C, αGalCer was efficiently presented by ILC3s as defined by

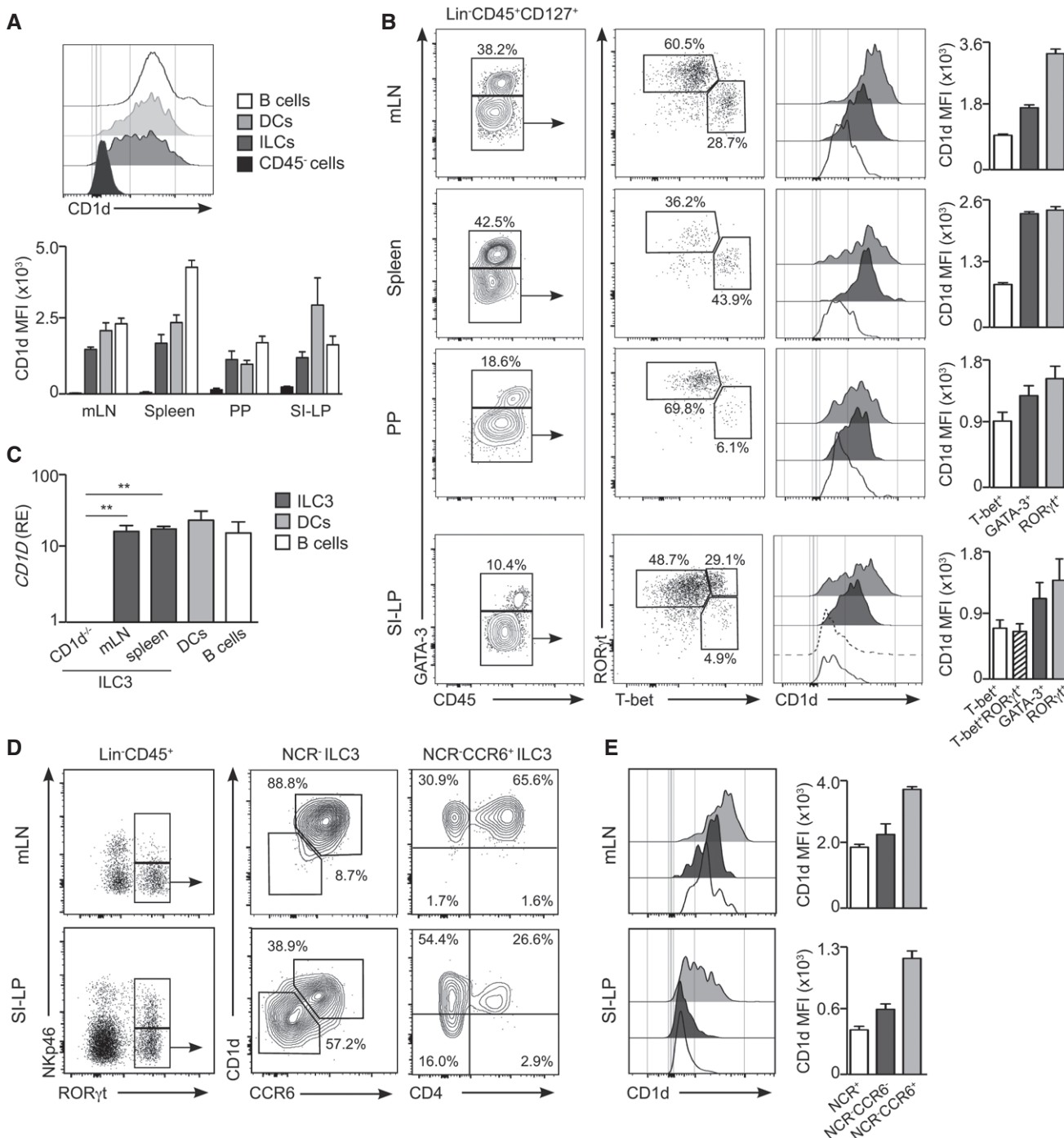

**Figure 1.  CD1d expression on ILCs.**

A   Flow cytometry for CD1d expression (top) and CD1d mean fluorescence intensity (MFI; bottom) on CD45⁻ cells, ILCs (Lin⁻CD45⁺CD127⁺), DCs (CD11c⁺) and B cells (B220⁺) of the depicted tissues from WT mice (n = 4).

B   Flow cytometry profiles showing gating strategy (dot plots), CD1d expression (histograms) and CD1d MFI (bars) in T-bet⁺ (empty profile), GATA-3⁺ (dark grey), RORγt⁺ (light grey) and T-bet⁺RORγt⁺ (dotted line) ILCs in mLN, spleen, PP and SI-LP (n = 5). Numbers indicate percentage of cells in the depicted gates.

C   Quantitative RT–PCR analysis of mRNA encoding CD1d in freshly isolated ILC3s (from mLN of CD1d-deficient mice and mLN and spleen of WT mice), B cells and DCs (n = 3). Results are normalized to those of *GAPDH*; RE, relative expression. **P < 0.01 (two-tailed unpaired *t*-test).

D   Flow cytometry profiles showing NKp46 and RORγt expression in Lin⁻CD45⁺ cells; CD1d and CCR6 expression in NCR⁻ ILC3s; and CD1d and CD4 expression in NCR⁻CCR6⁺ ILC3s from mLN and SI-LP (n = 4). Numbers indicate percentage of cells in the depicted gates.

E   CD1d expression (histograms) and CD1d MFI in NCR⁺ (empty profile), NCR⁻CCR6⁻ (dark grey) and NCR⁻CCR6⁺ (light grey) ILC3s from mLN and SI-LP (n = 4).

Data information: Graphs represent mean ± SEM. Data are from three (A, C–E) or five (B) independent experiments.

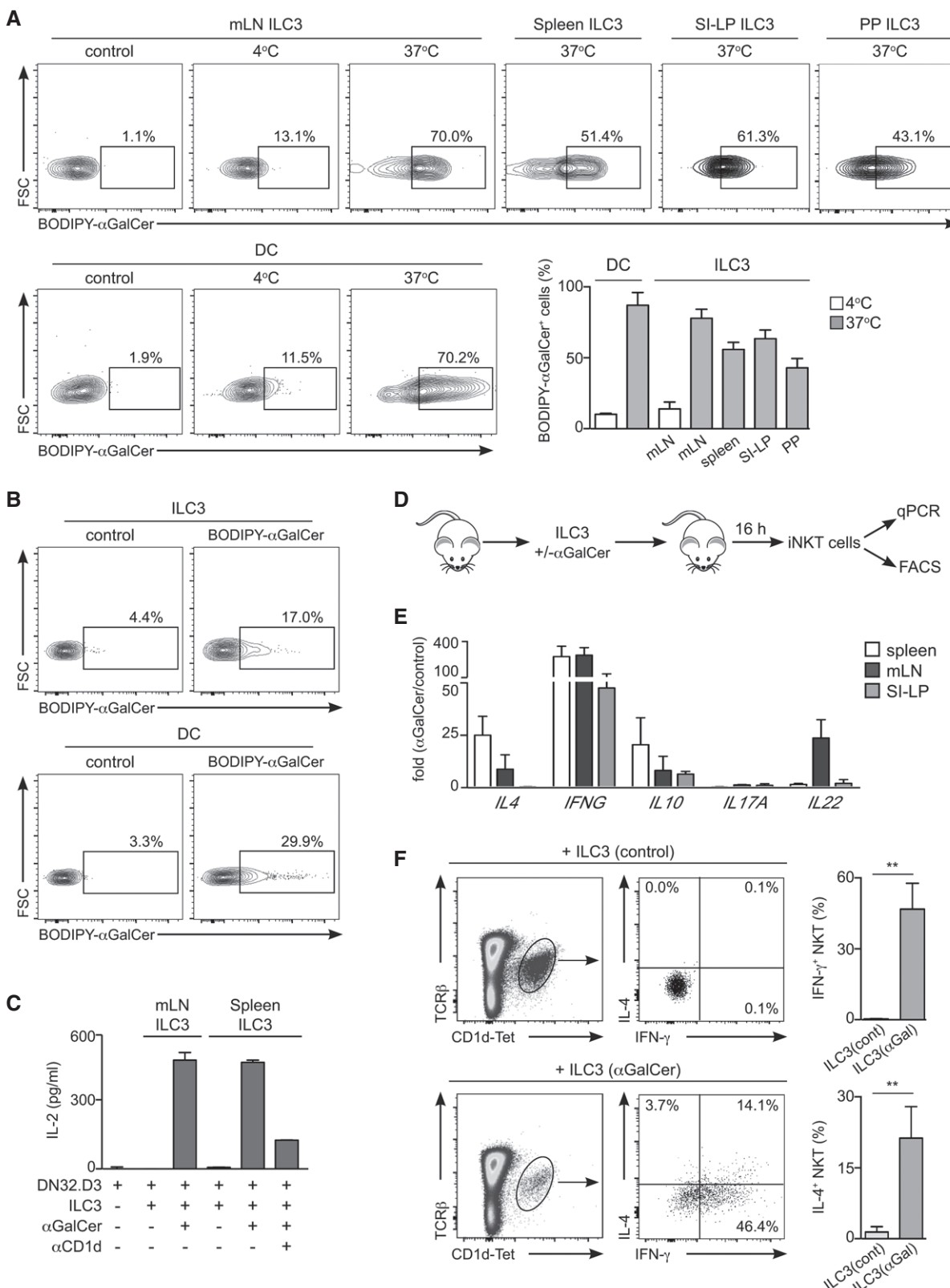

**Figure 2.**

IL-2 production by iNKT cells. Importantly, lipid presentation was dependent on CD1d as IL-2 secretion was strongly reduced in the presence of a CD1d-blocking antibody (Fig 2C). To further test whether ILC3s could mediate CD1d-dependent lipid presentation

**Figure 2. ILC3s internalize and present lipid antigens to iNKT cells.**

A    Cells were cultured in the presence or absence of BODIPY-αGalCer at 4 or 37°C as indicated. Flow cytometry profiles and percentages of BODIPY-αGalCer$^+$ DC or ILC3 from the indicated tissues incubated at 4°C (white bars) or 37°C (grey bars) are shown (*n* = 3).

B    WT mice were intravenously injected with BODIPY-αGalCer or PBS (control); lipid uptake was analysed by flow cytometry in the depicted populations from the spleen 16 h after lipid injection (*n* = 3). Numbers indicate percentage of cells in the depicted gates.

C    IL-2 secretion by DN32.D3 cells co-cultured with ILC3 sort-purified from mLN or spleen as indicated, pulsed (+) or not (−) with αGalCer and with or without αCD1d-blocking antibody.

D–F  Sort-purified ILC3s were preincubated with αGalCer (ILC3(αGal)) or PBS (ILC3(cont)) and adoptively transferred into WT recipients. (D) Experimental protocol; (E) fold change in mRNA expression for the indicated cytokines in iNKT cells sort-purified from spleen, mLN or SI-LP of mice injected with αGalCer-loaded ILC3s. Gene expression was measured by qPCR, normalized to *GAPDH* and presented as expression relative to iNKT cells sort-purified from mice injected with control ILC3s (*n* = 3–5). (F) Intracellular cytokine staining (left) and percentage of cytokine$^+$ cells (right) in splenic iNKT cells from ILC3-recipient mice (*n* = 3). Numbers indicate percentage of cells in the depicted gates. **$P < 0.01$, two-tailed unpaired *t*-test.

Data information: Graphs represent mean ± SEM. Data are from more than two independent experiments.

*in vivo*, we loaded sort-purified ILC3s with lipids (or PBS as control), transferred them into WT recipients and measured activation of iNKT cells sort-purified from spleen, mLN and SI-LP of the recipient mice (Fig 2D–F). Lipid-loaded ILC3s induced activation of iNKT cells as evidenced by an increase in the expression of cytokines detected by qPCR (Fig 2E). Consistent with these data, intracellular stainings showed increased IFN-γ and IL-4 production by splenic iNKT cells from mice receiving lipid-loaded ILC3s (Fig 2F).

Altogether our data demonstrate that ILC3s can internalize lipids, load them on CD1d and present them to iNKT cells both *in vitro* and *in vivo*. The capacity of ILC3s to mediate CD1d-dependent lipid presentation points towards an ILC3-iNKT cell dialogue that could modulate the initiation and/or progression of the immune response. It is worth noting that in the adult spleen, both ILC3 and iNKT cells are located in the splenic marginal zone [28,29], which may favour their crosstalk and the early initiation of immune responses. Within the intestine, iNKT cells function and features are modulated by CD1d-dependent recognition of commensal-derived lipids [33–35]. Since ILC3s modulate CD4$^+$ T-cell responses to commensal bacteria [3,8,9], it is conceivable that they could internalize and present commensal-derived lipids to iNKT cells consequently regulating intestinal iNKT cell immunity. Further studies will allow defining the functional consequences of ILC3-mediated lipid presentation *in vivo* both in homeostasis and during inflammatory and infectious responses.

### Engagement of CD1d on ILC3s drives IL-22 secretion

Next, we investigated the consequences of CD1d expression for ILCs' homeostasis and function. First, we tested whether CD1d expression and/or NKT cells could influence the ILC populations in steady state. T cells are known to regulate ILC numbers and function [7,36] and ILC populations are altered in RAG$^{-/-}$, TCRα$^{-/-}$ and ZAP70$^{-/-}$ mice that show a marked increase in the percentage of ILC2s in their mLN at the expense of ILC3s [7]. Likewise, NKT cells regulate immune cells in steady state including memory CD8$^+$ T cells and IgE production by B cells [37]. To determine whether NKT cells/CD1d regulate ILC development, we examined ILCs in CD1d-deficient mice (which also lack NKT cells; Figs 3A and B, and EV4). We found that ILC populations were comparable in CD1d$^{-/-}$ mice and WT littermates, suggesting that neither CD1d nor NKT cells play a major role in the regulation of ILC development and/or homeostasis (Figs 3A and B, and EV4).

We then explored whether CD1d expression can shape the function of ILC3s. Previous studies have shown that engagement of CD1d results in secretion of cytokines by both professional and unconventional APCs including IL-10 secreted by IEC and IL-12 by DCs and monocytes [21,22]. Consequently, engagement of epithelial CD1d renders protective effects in mouse models of IBD, while CD1d-mediated production of IL-12 by APCs leads to protection against viral infection [14,23]. To study whether CD1d could regulate ILC3 function, we performed antibody-mediated cross-linking of CD1d on ILC3s and measured changes in the expression of *IL22*, *IL17A*, *LTA* and *LTB* (cytokines typically produced by ILC3s) as well as *TNFSF13B* (BAFF), *TNFSF13* (APRIL) and *CD40LG* (CD40L; factors by which ILC3s modulate B-cell function [28]; Figs 3C and EV5). Strikingly, CD1d ligation on ILC3s resulted in a prominent increase in the expression of *IL22* mRNA (Fig 3C and D), suggesting that CD1d engagement is sufficient to induce cytokine production by ILC3s. Consistent with these data, we detected increased IL-22 production by ILC3s after CD1d cross-link by ELISA and intracellular staining (Fig 3E–G). ILC3s are known to produce IL-22 in response to a variety of environmental cues including cytokines (i.e. IL-23) or Toll-like receptor (TLR) ligands; and combinations of such stimuli may skew ILC3 activation and differentiation [4,38]. Interestingly, we observed that CD1d engagement on ILC3s acts synergistically with IL-23 to induce increased IL-22 production (Fig 3D–G). It seems likely that in the context of an immune response CD1d signals could function together with other factors (i.e. IL-23, IL-1β, TLR ligands) to modulate ILC3 activation and subsequently control the local cytokine milieu to maintain homeostasis and/or regulate inflammatory or infectious responses. Although the molecular mechanisms by which CD1d engagement modulates IL-22 production will require further investigation, it has been recently shown that CD1d engagement in IEC leads to STAT3 phosphorylation and STAT3-dependent cytokine secretion [14]. Interestingly, activated STAT3 can directly bind the *IL22* locus on ILC3s controlling IL-22 production [39], which may suggest STAT proteins as a possible mechanistic link between CD1d signalling and IL-22 secretion.

Finally, we tested whether ILC3s could participate in CD1d-mediated immune responses *in vivo*. To explore this, we administered αGalCer to WT mice, either intravenously or orally, and analysed changes on the features of ILC3s in response to lipid administration in the spleen or SI-LP, respectively (Fig 3H–J). We detected activation of splenic ILC3, evidenced by CD69 up-regulation, as early as 6 h after intravenous lipid injection (Fig 3H). Moreover, and in agreement with our *in vitro* data, ILC3s sorted from

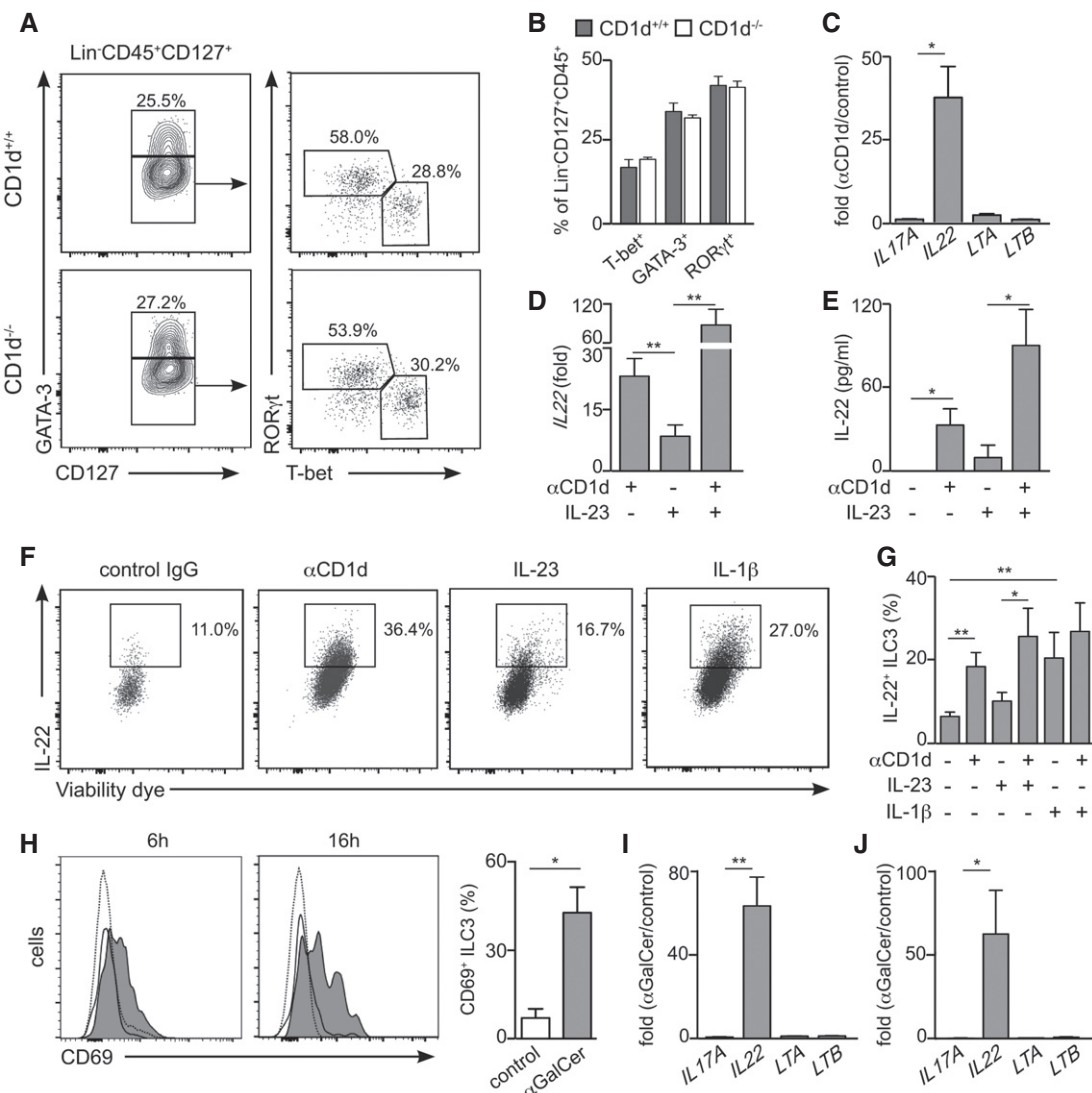

**Figure 3. Engagement of CD1d on ILC3s induces IL-22 production.**

A, B Flow cytometry plots showing gating strategy (A) and percentage of ILC populations (from Lin⁻CD45⁺CD127⁺ cells, B) in mLN from WT (*n* = 4) and CD1d-deficient (*n* = 3) mice.

C Fold change in mRNA expression for the indicated cytokines in ILC3s after antibody-mediated CD1d cross-link (*n* = 3). Gene expression was measured by qPCR and normalized to *GAPDH* and to the mRNA expression levels in control ILC3s. *$P < 0.05$ two-tailed unpaired *t*-test.

D Fold change in *IL22* mRNA expression in ILC3s after antibody-mediated CD1d cross-link and/or IL-23 stimulation (*n* = 4). Gene expression was measured by qPCR and normalized to *GAPDH* and to the mRNA expression levels in control ILC3s. **$P < 0.01$ two-tailed unpaired *t*-test.

E IL-22 detection by ELISA in the supernatant of ILC3s cultured in the presence of αCD1d and/or IL-23 (*n* = 3). *$P < 0.05$ two-tailed unpaired *t*-test.

F, G Intracellular cytokine staining (F) and percentage of IL-22⁺ ILC3s (G) after stimulation with αCD1d, IL-23 and/or IL-1β. *$P < 0.05$, **$P < 0.01$ two-tailed unpaired *t*-test (*n* = 4–8).

H WT mice were i.v. injected with αGalCer or PBS (control) and activation of splenic ILC3s was assessed by flow cytometry as CD69 up-regulation at 6 or 16 h after injection. Grey filled profile, ILC3 from αGalCer-injected mice; empty profile, ILC3 from PBS-injected mice; dotted line, CD45⁻ cells. Right graph, percentage of CD69⁺ ILC3s 6 h after injection of αGalCer (grey) or PBS (white) (*n* = 3). *$P < 0.05$ two-tailed unpaired *t*-test.

I, J Fold change in mRNA expression for the indicated cytokines in ILC3s sort-purified from spleen (I) or SI-LP (J) 6 h after intravenous (I) or oral (J) αGalCer administration. Gene expression was measured by qPCR and normalized to *GAPDH* and to the expression levels in ILC3s sort-purified from PBS-injected mice (*n* = 3). *$P < 0.05$, **$P < 0.01$ two-tailed unpaired *t*-test.

Data information: Graphs represent mean ± SEM.

αGalCer-injected mice showed a marked increase in the expression of *IL22* in comparison with ILC3s from control mice (Fig 3I). Also, oral administration of αGalCer resulted in increased *IL22* production by intestinal ILC3s (Fig 3J), in line with a functional role for ILC3s

in the regulation of lipid-mediated immunity within the intestine. Although IL-22 generally supports epithelial barrier function and tissue repair, it can also promote proinflammatory responses and its levels are often high in chronic inflammatory conditions [40].

 

Moreover, IL-22 contributes to the control of bacterial and fungal infection, and to protective barrier functions during viral infections [41]. Since lipid-dependent activation of immune cells modulates the outcome of a variety of infectious and inflammatory diseases [10,11,14,16], it is likely that ILC3s contribute to lipid-mediated immunity in those contexts through secretion of IL-22, consequently shaping the initiation or development of the immune response.

All together, our data identify a novel pathway for immune regulation mediated by CD1d expression on ILC3s. We show that ILC3s can internalize and present lipids on CD1d and that CD1d-dependent activation of ILC3s induces secretion of IL-22. This pathway could have implications for a variety of immune responses where CD1d-dependent immunity plays a central role [10,11]. For instance, CD1d contributes to the functional regulation of immune cells in a variety of autoimmune disorders (such as IBD or systemic lupus erythematosus) consequently shaping the initiation and/or progression of the immune response [42–45]. Accordingly, CD1d expression is down-regulated on IECs from patients with IBD and on B cells from patients with systemic lupus erythematosus, which correlates with abnormal function of IECs, B cells and NKT cells in these diseases. Thus, similarly to the protective role exerted by epithelial CD1d in intestinal inflammation by controlling IL-10 secretion, CD1d on ILC3s may contribute to the regulation of immunity by modulating the production of IL-22. Analyses of ILCs in human tissues will be crucial to define the expression of the different CD1 isoforms (CD1a-d) in the ILC families and therefore identify their role in the regulation of iNKT cells in homeostasis and disease. Further studies will allow defining the mechanisms that control CD1d expression on ILC3s and the role of the ILC-NKT cell axis in different diseases providing a novel target for immune intervention.

# Materials and Methods

### Reagents and mice

C57BL/6 WT mice were purchased from Charles River and CD1d$^{-/-}$ mice were bred at King's College London. All experiments were approved by the King's College London's Animal Welfare and Ethical Review Body and the United Kingdom Home Office (project licence 70/7907). iNKT hybridoma DN32.D3 was from A. Bendelac (University of Chicago). αGalCer was purchased from BioVision. PBS57-loaded CD1d tetramers were provided by the NIH Tetramer Core Facility. BODIPY-αGalCer was synthesized according to previously published protocols [46].

### Tissue preparation and flow cytometry

Tissues were processed as previously described [3] and single-cell suspensions were used for flow cytometry. For SI-LP lymphocyte preparations, intestines were isolated, PP removed, and intestines were flushed with cold PBS and cut open longitudinally. Tissues were incubated 20 min at 37°C in HBSS, 1 mM EDTA, 5% FCS to remove epithelial cells and intraepithelial lymphocytes. The lamina propria layer was isolated after incubation for 40 min at 37°C with collagenase (1.5 mg/ml) and DNase (100 µg/ml). Spleen and mLN

were harvested and single-cell suspensions obtained by mincing the tissue through a 45-µm strainer.

Flow cytometry analyses were performed in FACS buffer containing PBS, 1% BSA and 1% FCS. Anti-mouse CD16/32 (clone 2.4G2, Biolegend) was used to block non-specific antibody binding. Afterwards cells were stained with the following antibodies, all from Biolegend unless specified otherwise: B220 (clone RA3-6B2), CCR6 (clone 29-2L17), CD117 (clone 2B8), CD11b (clone M1/70), CD11c (clone N418), CD127 (clone A7R34), CD1d (clone 1B1), CD3 (clone 145-2C11), CD4 (clone GK1.5), CD45.2 (clone 104), CD5 (clone 53-7.3), CD69 (clone H1.2F3), CD90 (clone 53-2.1), I-A/I-E (clone M5-114.15.2), NKp46 (clone 29A1.4), TCR-β (clone H57-587), sca-1 (clone, D7), T-bet (clone 4B10), PLZF (clone 9E12), GATA-3 (clone TWAJ, eBioscience), RORγt (clone B2D, eBioscience).

For intracellular staining, cells were fixed and permeabilized with Foxp3/Transcription Factor Staining Buffer Set (eBioscience). Dead cells were excluded from the analyses using a fixable viability stain (Biolegend). Flow cytometry data were collected on a FACS-Canto II flow cytometer (BD Biosciences) and were analysed with FlowJo software (Tree Star).

### Isolation, culture and stimulation of ILC3s

For purification of ILC3s, single-cell suspensions from mLNs and spleens were prepared by mincing the tissue through a 45-µm strainer. Splenic ILC3s were enriched by using lineage cell depletion kit (Miltenyi Biotec). ILC3s were further purified by cell sorting as CD45$^{+}$CD127$^{+}$CD117$^{+}$B220$^{-}$CD3$^{-}$CD5$^{-}$CD11c$^{-}$CD11b$^{-}$ cells [3,9,28] with a FACSAria II (BD Biosciences). Sorted ILC3s were cultured in complete RPMI medium supplemented with 10% FCS. For cross-linking experiments, ILCs were expanded for 5–8 days with IL-7 (50 ng/ml) and IL-2 (100 ng/ml) as described [28,47].

Surface CD1d was cross-linked by incubation of ILC3s with anti-CD1d antibody (clone 1B1) followed by goat anti-rat IgG as described [14,21]. In some experiments, cells were incubated with recombinant IL-23 (100 ng/ml, Biolegend) or IL-1β (100 ng/ml, Biolegend) and/or anti-CD1d. For intracellular cytokine staining after cross-linking, ILC3s were incubated with brefeldin A (5 µg/ml, Biolegend) for the final 2 h of stimulation, fixed and permeabilized with Fix/Perm buffer set (Biolegend) and stained for cytokine production with anti-mouse IL-22 (1H8PWSR, eBioscience). Dead cells were excluded from the analyses using a fixable viability stain (Biolegend). IL-22 was measured in the cultures' supernatant using IL-22 ELISA MAX Deluxe (Biolegend).

For analysis of BODIPY-αGalCer acquisition, ILC3s were incubated with fluorescent lipids (100 ng/ml) for 4–6 h before flow cytometry analyses. For analysis of fluorescent lipid uptake *in vivo*, mice were intravenously injected with BODIPY-αGalCer (1 µg/mouse) in 200 µl of PBS. Lipid uptake was analysed by flow cytometry in the spleen 16 h after injection.

For αGalCer presentation analysis, sorted cells were incubated for 2 h with αGalCer (100 ng/ml), washed extensively and cultured (5 × 10$^3$ cells/well) with 2 × 10$^4$ cells/well of DN32.D3 cells for 16 h. IL-2 concentration was determined by a standard sandwich ELISA in the supernatant of the culture medium, using anti-IL-2 (JES6-1A12) antibody for capture and biotinylated anti-IL-2 (JES6-5H4) antibody for detection (both from Biolegend).

## ILC3 adoptive transfer and iNKT cell activation

ILC3s were incubated for 2 h with αGalCer (100 ng/ml) or PBS, washed extensively and adoptively transferred into WT recipients ($0.5–1 \times 10^5$ cells/mouse). Recipient mice were euthanized 18–24 h following transfer and iNKT cell activation was measured by qPCR and flow cytometry.

For qPCR analyses, iNKT cells were sort-purified from the spleen, mLN and SI-LP of recipient mice as TCR-$\beta^+$CD1d-tetramer$^+$B220$^-$ cells and RNA was extracted as described below.

For intracellular IFN-$\gamma$ and IL-4 staining, single-cell suspension were prepared and stained with B220, TCR-$\beta$ and CD1d-tetramer prior to fixation and permeabilization with Fix/Perm buffer set (Biolegend). Fixed cells were stained with anti-mouse IFN-$\gamma$ (XMG1.2, Biolegend) and anti-IL-4 (11B11, Biolegend). Dead cells were excluded from the analyses using a fixable viability stain (Biolegend).

## Quantitative real-time PCR

RNA samples were prepared using RNeasy Mini Kit (Qiagen) and cDNAs were synthesized with iScript Select cDNA Synthesis Kit (Bio-Rad). For gene expression analyses, we used SYBR green Master kit (Bio-Rad) and the following primers: *GAPDH-F*: 5′-ACGA CCCCTTCATTGAC-3′; *GAPDH-R*: 5′-TCCACGACATACTCAGCAC-3′; *LTA-F*: 5′-GCCATTCCCACTCCCATCTACCTG-3′; *LTA-R*: 5′-CGCACC CACGGTCCTTGAAGTC-3′; *LTB-F*: 5′-ACCTCATAGGCGCTTGGAT G-3′; *LTB-R*: 5′-ACGCTTCTTCTTGGCTCGC-3′; *IL17A-F*: 5′-AGCAAG AGATCCTGGTCCTGAA-3′; *IL17A-R*: 5′-CATCTTCTCGACCCTGAAA GTGA-3′; *IL22-F*: 5′-GACCAAACTCAGCAATCAGCTC-3′; *IL22-R*: 5′-TACAGACGCAAGCATTTCTCAG-3′; *CD40L-F*: 5′-AGCTGGTGCTT CTGTGTTTGTC-3′; *CD40L-R*: 5′-AAGATGAGAAGCCAACTCTGTG G-3′; *TNFSF13-F*: 5′-ACCCAGAAGCACAAGAAGAAGC-3′; *TNFSF13-R*: 5′-GTACTGGTTGCCACATCACCTC-3′; *TNFSF13B-F*: 5′-AGCTGAGCC TGGTGACCCTGTTC-3′; *TNFSF13B-R*: 5′-TCTCATCTCCTTCTTCCAG CCTCGC-3′; *IL4-F*: 5′-AAGAACACCACAGAGAGTGAGCTC-3′; *IL4-R*: 5′-TTTCAGTGATGTGGACTTGGACTC-3′; *IL10-F*: 5′-AGAAGCATGG CCCTGAAATCAAGG-3′; *IL-10-R*: 5′-CTTGTAGACACCTTGGTCTTG GAG-3′; *IFNG-F*: 5′-GCCATCAGCAACAACATAAGCGTC-3′; *IFNG-R*: 5′-CCACTCGGATGAGCTCATTGAATG-3′; *CD1D-F*: 5′-GAATGACACCT GCCCCCTATT-3′; *CD1D-R:* 5′-ACCCACACAGGTTTTGGGTA-3′. Reactions were run in a real-time PCR system (ABI7900HT; Applied Biosystems).

**Expanded View** for this article is available online.

## Acknowledgements
This work was funded by the Medical Research Council (grants to P.B. MR/L008157/1; and to G.S.B. G1001750); J.S.d.G. and R.J. were supported by Marie Curie Intra-European Fellowships (PIEF-GA-2013-627391 and H2020-MSCA-IF-2015-703639). We acknowledge the NIH Tetramer Core Facility (contract HHSN272201300006C) for provision of CD1d tetramers. We thank J. Spencer and S. John for critical reading of the manuscript.

## Author contributions
JSdG and RJ designed and performed research, analysed data and wrote the manuscript; NF designed and performed research; PJJ, LRC and GSB provided critical reagents and contributed to design and editing of the manuscript; PB supervised and designed research, analysed data and wrote the manuscript.

## Conflict of interest
The authors declare that they have no conflict of interest.

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
