## [Review Process File · EMBO Reports]

Manuscript EMBO-2016-42412

CD1d-MEDIATED ACTIVATION OF GROUP 3 INNATE LYMPHOID CELLS DRIVES IL-22 PRODUCTION

Julia Saez de Guinoa, Rebeca Jimeno, Nazanin Farhadi, Peter J Jervis, Liam R Cox, Gurdyal S Besra, Patricia Barral

Corresponding author: Patricia Barral, King's College London

Review timeline:

Submission date:	21 March 2016
Editorial Decision:	21 April 2016
Revision received:	24 June 2016
Editorial Decision:	15 July 2016
Revision received:	07 September 2016
Accepted:	04 October 2016

Editor: Achim Breiling

Transaction Report:

1st Editorial Decision

21 April 2016

Thank you for your submission to EMBO reports. We have now received reports from the three referees that were asked to evaluate your study, which can be found at the end of this email. As you will see, although the referees find the topic of interest, they raise several concerns and suggest additional experiments to strengthen the conclusions of the study.

As the reports are below, I will not detail them here. However, it is most important to provide in vivo evidence that ILC3s are necessary for iNKT activation (point 3 of referee #1 and point 4 by referee #4). Further, points 1 and 2 of referee #1 deserve particular attention (functional studies in the gut), as well as point 5 of referee #3 (CD1 RNA expression in ILC3s).

Given these constructive comments, we would like to invite you to revise your manuscript with the understanding that all referee concerns (as detailed in their reports) must be fully addressed in a complete point-by-point response. Acceptance of the manuscript will depend on a positive outcome of a second round of review. It is EMBO reports policy to allow a single round of revision only and acceptance or rejection of the manuscript will therefore depend on the completeness of your responses included in the next, final version of the manuscript.

Revised manuscripts should be submitted within three months of a request for revision; they will otherwise be treated as new submissions. Please contact us if a 3-months time frame is not sufficient for the revisions so that we can discuss the revisions further.

Supplementary/additional data: The Expanded View format, which will be displayed in the main

HTML of the paper in a collapsible format, has replaced the Supplementary information. You can submit up to 5 images as Expanded View. Please follow the nomenclature Figure EV1, Figure EV2 etc. The figure legend for these should be included in the main manuscript document file in a section called Expanded View Figure Legends after the main Figure Legends section. Additional Supplementary material should be supplied as a single pdf labeled Appendix. The Appendix includes a table of content on the first page, all figures and their legends. Please follow the nomenclature Appendix Figure Sx throughout the text and also label the figures according to this nomenclature. For more details please refer to our guide to authors.

Important: All materials and methods should be included in the main manuscript file.

Regarding data quantification and statistics, can you please specify the number "n" for how many experiments were performed, the bars and error bars (e.g. SEM, SD) and the test used to calculate p-values in the respective figure legends? This information must be provided in the figure legends. Please also include scale bars in all microscopy images.

We now strongly encourage the publication of original source data with the aim of making primary data more accessible and transparent to the reader. The source data will be published in a separate source data file online along with the accepted manuscript and will be linked to the relevant figure. If you would like to use this opportunity, please submit the source data (for example scans of entire gels or blots, data points of graphs in an excel sheet, additional images, etc.) of your key experiments together with the revised manuscript. Please include size markers for scans of entire gels, label the scans with figure and panel number, and send one PDF file per figure or per figure panel.

- a complete author checklist, which you can download from our author guidelines (<http://embor.embopress.org/authorguide#revision>). Please insert page numbers in the checklist to indicate where the requested information can be found.
- a letter detailing your responses to the referee comments in Word format (.doc)
- a Microsoft Word file (.doc) of the revised manuscript text
- editable TIFF or EPS-formatted figure files in high resolution

Please note that we now mandate that all corresponding authors list an ORCID digital identifier!

As part of the EMBO publication's Transparent Editorial Process, EMBO reports publishes online a Review Process File to accompany accepted manuscripts. This File will be published in conjunction with your paper and will include the referee reports, your point-by-point response and all pertinent correspondence relating to the manuscript.

I look forward to seeing a revised version of your manuscript when it is ready. Please let me know if you have questions or comments regarding the revision.

REFeree REPORTS

Referee #1:

The paper by Saez de Guinoa et al, shows that ILC are the immune cells that express the highest levels of CD1d, a lipid-presenting molecule which preferentially activates iNKT cells. Within ILC, CCR6+ ILC3 express the highest levels of CD1d and when triggered in vitro and in vivo, these cells release IL-22, a cytokine necessary for general epithelial barrier functions. This paper demonstrates

that CCR6+RORgt ILC3 express CD1d, which represents a new layer ILC3 function in the control of homeostasis and inflammation. This is novel and adds to the understanding of ILC biology. Experiments are correctly executed, figures are clear and it reads well, even though the authors should consider shortening the three first paragraphs in one or two and make one figure out of figures 1 and 2 as described below.

However, there are issues that make this manuscript less exciting that it could have been.

1. Figures 1 and 2 could be made in one single figure that could be more straight forward with the suggestions below:

Figure 1 A: even though interesting, expression of CD1d on all ILC is not so relevant, since different ILCs exert different roles, which are tissue-dependent. A discrimination between different populations of ILC is nowadays indispensable. Figure 1A could be put in supp material.

In Figure 1B, C and D, it would be helpful to have error bars on graphs. Even though data are representative, it would be nice to see the heterogeneity of CD1d expression within each population. Figure 1D : Tbet staining is a difficult one and it is therefore difficult to separate populations. The authors should perform the classical NKp46/RORgt staining to discriminate between NCR+, NCR- ILC3 and ILC1. Tbet+RORgt+ cells comprise ILC that are transitioning between ILC3 and ILC1 like the authors mention, but also and importantly, NCR+ ILC3, which should be precised in the text. Therefore the Tbet/RORgt staining might not be the best approach to look at several ILC populations. NKp46/RORgt staining, then gating on NCR- cells and looking at CCR6 and CD4 expression to isolate LTi cells would be best to replace Fig 1D, E and F. Accordingly, in the text, the paragraph about ILC3 populations comes a bit late, and could be integrated in the previous paragraph.

2. The lack of gut ILC3 functional studies is the biggest caveat of the paper. ILC3 present several features depending on the organs, and tissue microenvironment is likely to play a major role in the function of ILC in general. Since ILC3 are mostly present in the gut, it would be interesting to look at their function in this organ (where ILC3 might also be more likely to acquire lipid antigens under physiological conditions). Moreover, the authors nicely show the increased IL-22 secretion by ILC3 after crosslinking of CD1d, and IL-22 is particularly important to maintain homeostasis in the gut. Therefore, the authors should look at SI, colonic and Peyer's patches ILC3 (most likely CCR6+ ILC3) and if these gut ILC3 are also able to acquire lipid antigens in vitro and in vivo, as the authors did for mLN and splenic ILC3 in figure 3.

In that case, iv injection might not be the most physiological way to give lipids to mice. Is it possible to add traceable (fluorescent ?) lipid in the food or gavage mice with fluorescent α GalCer to reach gut ILC3 ?

3. The crosstalk between ILC3 and iNKT cells is the most interesting finding of this manuscript. Therefore, the response of iNKT cells after presentation of lipids through CD1d on ILC3 should be more extensively studied. In figure 3C, authors should show other cytokines such as IL-4, IL-13, IL-17, IFN γ , IL-10 (produced by iNKT in adipose tissue). It would be interesting to see if ILC3 from different organs « instruct » iNKT cells in similar or different ways (gut vs mLN vs spleen). For example, gut ILC3 might induce more of a tolerogenic iNKT phenotype than mLN or splenic ILC3.

In the same vein, cocultures of iNKT and ILC3 could be used to look at ILC3 function, since it would be more physiological than antibody-mediated crosslinking of CD1d.

Furthermore, in the in vivo experiments, the authors nicely show the activation of ILC3 and increased IL-22 production after α GalCer administration. The authors should look at the phenotype of iNKT cells in these same experiments (secreted cytokines (such as cited above) by FACS or qPCR).

Other comments

- The authors should be aware that indeed, all LTi are CCR6+, but not all RORgt+CCR6+ are LTi.
- Figure 2C (qPCR of lymphotoxin alpha) is not per se relevant for the paper and could be put in an additional figure.

Referee #2:

In this manuscript, Saez de Guinoa et al. report the identification of a novel IL-22-inducing immune pathway involving CD1d expression by group 3 innate lymphoid cells (ILC3s). Though expressed by different types of ILCs from a variety of tissues, CD1d was found to be particularly elevated on the surface of mucosal CCR6+NKp46-ILC3s. In vitro experiments demonstrated that these ILC3s captured lipids through CD1d and thereafter induced CD1d-dependent lipid presentation to invariant NKT (iNKT) cells. Additional in vitro and in vivo experiments showed that engagement of CD1d on ILC3s induced production of IL-22, a cytokine enhancing epithelial integrity and mucosal homeostasis. It is concluded that ILC3s mediate CD1d-mediated immune responses.

General comment

This is a clearly written and elegantly presented manuscript that shows a novel aspect of ILC biology. The main conclusion is strongly supported by compelling results obtained through well-performed experiments. The following additional comments are provided to further enhance this study.

Specific comments

- 1) In this manuscript, all the data were obtained by analyzing ILCs from mouse tissues. To enhance the overall impact and physiological relevance of this work, it would be important to show CD1d expression on human ILCs following gating strategies proposed by Hazenberg and Spits (Blood. 2014, 124:700-709).
- 2) Besides enhancing the integrity of mucosal epithelial cells, ILC3s collaborate with T and B cells to support homeostatic immune responses in both gut and spleen. These responses involve ILC expression of MHC-II and B cell-stimulating factors, including CD40L, BAFF and APRIL. Does CD1d engagement up-regulate ILC3 expression of these molecules?
- 3) Do CD1d-positive ILCs express Plzf, a transcription factor encoded by Bztb16 that supports defensive programs in CD1d-restricted NKT cells?

 Referee #3:

In this report, De Guinoa et al show that mouse ILC3 express CD1d, present lipid antigens to invariant NKT cells (iNKT) and can be stimulated to produce IL-22 through CD1d engagement. The report is interesting but there are several flaws that diminish my enthusiasm for the study

Major concerns

1. The authors show that ILC3 can uptake lipid antigens in vitro and in vivo. It is unclear whether this is a real function of ILC3 or simply reflects a non-specific uptake of lipids. The authors show uptake by DC and ILC3. They should show whether other cells, like B cells and T cells, endocytose lipids to a similar extent in the same conditions or whether this is a unique feature of ILC3 and DC.
2. Which receptors mediate internalization of lipids in ILC3? No molecular basis for this phenomenon is provided
3. The authors propose that engagement of CD1d with antibody-mediated cross-linking induces IL-22. The intracellular increase of IL-22 is rather unimpressive. It is also a mystery how this can occur. IL-22 production is dependent on STAT signaling. Which signaling pathway is induced by CD1d? How do those signal activate the IL-22 promoter? Provided that anti-CD1d augments IL-22 secretion, it seems more likely that the antibody activates Fc receptors or acts through other mechanisms
4. Lipid loaded ILC3 activate iNKT in vitro. However, the authors don't provide any evidence that ILC3 are necessary for the activation of iNKT in vivo.
5. The authors show no RNA expression for CD1d. According to the available mRNA data from Immgen, CD1d RNA is almost undetectable in ILC3, suggesting that CD1d expression may be related to cross-dressing. The authors should address this possibility by directly measuring CD1d RNA amount.

Reply to reviewers

We are very grateful for the positive and constructive comments provided by the reviewers and their thoughtful questions. We believe that the additional experiments performed have strengthened the manuscript and revealed important new insights into the role of ILCs in CD1-dependent immune responses

Referee #1 (R1)

The paper by Saez de Guinoa et al, shows that ILC are the immune cells that express the highest levels of CD1d, a lipid-presenting molecule which preferentially activates iNKT cells. Within ILC, CCR6+ ILC3 express the highest levels of CD1d and when triggered in vitro and in vivo, these cells release IL-22, a cytokine necessary for general epithelial barrier functions. This paper demonstrates that CCR6+RORgt ILC3 express CD1d, which represents a new layer ILC3 function in the control of homeostasis and inflammation. This is novel and adds to the understanding of ILC biology. Experiments are correctly executed, figures are clear and it reads well, even though the authors should consider shortening the three first paragraphs in one or two and make one figure out of figures 1 and 2 as described below.

We would like to thank R1 for their appreciation of the interest of our manuscript to the field and the contribution that it makes to the understanding of ILC biology.

R1 raised a number of specific concerns that we have addressed as follows:

1. Figures 1 and 2 could be made in one single figure that could be more straight forward with the suggestions below:

Figure 1 A: even though interesting, expression of CD1d on all ILC is not so relevant, since different ILCs exert different roles, which are tissue-dependent. A discrimination between different populations of ILC is nowadays indispensable. Figure 1A could be put in supp material. In Figure 1B, C and D, it would be helpful to have error bars on graphs. Even though data are representative, it would be nice to see the heterogeneity of CD1d expression within each population. Figure 1D : Tbet staining is a difficult one and it is therefore difficult to separate populations. The authors should perform the classical NKp46/RORgt staining to discriminate between NCR+, NCR- ILC3 and ILC1. Tbet+RORgt+ cells comprise ILC that are transitioning between ILC3 and ILC1 like the authors mention, but also and importantly, NCR+ ILC3, which should be precised in the text. Therefore the Tbet/RORgt staining might not be the best approach to look at several ILC populations. NKp46/RORgt staining, then gating on NCR- cells and looking at CCR6 and CD4 expression to isolate LTi cells would be best to replace Fig 1D, E and F. Accordingly, in the text, the paragraph about ILC3 populations comes a bit late, and could be integrated in the previous paragraph.

As suggested by R1 we have merged Figures 1 and 2 in a single figure (new Figure 1) and modified the text accordingly:

We have moved Figure 1A to the supplementary figures (now Figure EV1A)

We have included error bars for the CD1d MFI in all analyzed tissues in the revised Figures 1A, 1B and 1E

We have performed the NKp46/RORgt/CCR6/CD4 staining to differentiate NCR+ ILC3s, NCR- ILC3s and LTi cells. This new data has been included in the revised Figure 1D-E and discussed in the revised manuscript.

We have shortened the first 3 paragraphs in 2 as suggested, and moved forward the paragraph about ILC3 populations.

2. The lack of gut ILC3 functional studies is the biggest caveat of the paper. ILC3 present several features depending on the organs, and tissue microenvironment is likely to play a major role in the

function of ILC in general. Since ILC3 are mostly present in the gut, it would be interesting to look at their function in this organ (where ILC3 might also be more likely to acquire lipid antigens under physiological conditions). Moreover, the authors nicely show the increased IL-22 secretion by ILC3 after crosslinking of CD1d, and IL-22 is particularly important to maintain homeostasis in the gut. Therefore, the authors should look at SI, colonic and Peyer's patches ILC3 (most likely CCR6+ ILC3) and if these gut ILC3 are also able to acquire lipid antigens in vitro and in vivo, as the authors did for mLN and splenic ILC3 in figure 3. In that case, iv injection might not be the most physiological way to give lipids to mice. Is it possible to add traceable (fluorescent?) lipid in the food or gavage mice with fluorescent α GalCer to reach gut ILC3?

As suggested by R1 we have tested lipid uptake by intestinal ILC3s (from Peyer's patches and lamina propria) and showed that they can indeed acquire lipids *in vitro*. This new data has been included in Figure 2A.

We have further confirmed a role for intestinal ILC3s in lipid-mediated immunity *in vivo* by demonstrating that intestinal ILC3s produce IL-22 in response to oral administration of lipids. This new data has been included in Figure 3G.

We agree with R1 that visualization of lipid uptake in vivo after oral administration will be extremely interesting. We have orally gavaged mice with fluorescent α GalCer but, despite our best efforts, we have been unable to detect fluorescent lipids in the intestinal lamina propria (in any cell type). We believe that this is due to technical limitations as when arriving into the intestine lipids are probably too diluted and fluorescence signal is below the limit of detection of the flow-cytometer (note that based on the synthetic methodology each lipid molecule is labeled only with one BODIPY molecule; *Jervis et al, J Org Chem, 2011*).

3. The crosstalk between ILC3 and iNKT cells is the most interesting finding of this manuscript. Therefore, the response of iNKT cells after presentation of lipids through CD1d on ILC3 should be more extensively studied. In figure 3C, authors should show other cytokines such as IL-4, IL-13, IL-17, IFN γ , IL-10 (produced by iNKT in adipose tissue). It would be interesting to see if ILC3 from different organs « instruct » iNKT cells in similar or different ways (gut vs mLN vs spleen). For example, gut ILC3 might induce more of a tolerogenic iNKT phenotype than mLN or splenic ILC3. In the same vein, cocultures of iNKT and ILC3 could be used to look at ILC3 function, since it would be more physiological than antibody-mediated crosslinking of CD1d. Furthermore, in the in vivo experiments, the authors nicely show the activation of ILC3 and increased IL-22 production after α GalCer administration. The authors should look at the phenotype of iNKT cells in these same experiments (secreted cytokines (such as cited above) by FACS or qPCR).

We would like to point out that primary murine iNKT cells have extremely low viability *in vitro* (~90% of sorted cells die within 3h of culture) due to the high expression of the P2X7 ion channel (*Kawamura et al, J Immunol 2006; Seman et al, Immunity 2003; Aswad et al, Cell Immunol 2006*). To overcome this, we use the iNKT cell hybridoma DN32.D3 (*Lantz and Bendelac, J Exp Med 1994*) for our *in vitro* experiments, which is widely used in the field to study CD1d-dependent lipid presentation. Experiments aiming to characterize ILC3-primary iNKT cell crosstalk have been performed *in vivo* (see below).

As suggested by R1 we measured cytokine secretion by DN32.D3 iNKT cells after co-culture with ILC3s (IFN- γ , IL-4, IL-13, IL-17, IL-10); however we could only detect consistently high secretion of IL-2 which is the predominant cytokine produced by DN32.D3 cells.

We have now confirmed that ILC3s can mediate lipid presentation and activation of iNKT cells *in vivo* within different anatomical locations (Figure 2D-F). Moreover, as R1 suggested we have analyzed the effect of ILC3-mediated lipid presentation to iNKT cells in spleen, mLN and SI-LP and observed different patterns of cytokine secretion by iNKT cells in the different organs (as detected by qPCR and flow-cytometry). This new data is included in the new Figure 2D-F.

Other comments

- The authors should be aware that indeed, all LT α are CCR6+, but not all ROR γ t+CCR6+ are LT α . We thank the reviewer for his comment and have changed the text accordingly

- Figure 2C (qPCR of lymphotoxin alpha) is not per se relevant for the paper and could be put in an additional figure.

We agree with the reviewer and moved the panel to the supplementary figures (Figure EV1C)

Referee #2 (R2):

In this manuscript, Saez de Guinoa et al. report the identification of a novel IL-22-inducing immune pathway involving CD1d expression by group 3 innate lymphoid cells (ILC3s). Though expressed by different types of ILCs from a variety of tissues, CD1d was found to be particularly elevated on the surface of mucosal CCR6+NKp46-ILC3s. In vitro experiments demonstrated that these ILC3s captured lipids through CD1d and thereafter induced CD1d-dependent lipid presentation to invariant NKT (iNKT) cells. Additional in vitro and in vivo experiments showed that engagement of CD1d on ILC3s induced production of IL-22, a cytokine enhancing epithelial integrity and mucosal homeostasis. It is concluded that ILC3s mediate CD1d-mediated immune responses.

General comment

This is a clearly written and elegantly presented manuscript that shows a novel aspect of ILC biology. The main conclusion is strongly supported by compelling results obtained through well-performed experiments. The following additional comments are provided to further enhance this study.

We would like to thank R2 for their appreciation of the interest of our manuscript to the field and the contribution that it makes to the understanding of ILC biology.

Specific comments:

1) In this manuscript, all the data were obtained by analyzing ILCs from mouse tissues. To enhance the overall impact and physiological relevance of this work, it would be important to show CD1d expression on human ILCs following gating strategies proposed by Hazenberg and Spits (Blood. 2014, 124:700-709).

We agree with R2 that this will be an interesting experiment. Because both ILCs and iNKT cells are tissue-resident cells, and there are striking phenotypical differences in cells from different tissues, we are in the process of obtaining ethical approval to undertake extensive studies regarding CD1 expression on ILCs from human tissues.

2) Besides enhancing the integrity of mucosal epithelial cells, ILC3s collaborate with T and B cells to support homeostatic immune responses in both gut and spleen. These responses involve ILC expression of MHC-II and B cell-stimulating factors, including CD40L, BAFF and APRIL. Does CD1d engagement up-regulate ILC3 expression of these molecules?

We thank R2 for this suggestion. We have now performed additional experiments to test whether CD1d cross-link regulates the expression of the suggested molecules. The new data is now presented in new Figure EV5

3) Do CD1d-positive ILCs express Plzf, a transcription factor encoded by Bztb16 that supports defensive programs in CD1d-restricted NKT cells?

We have performed a PLZF staining in RORgt+ ILC3s and included this data in Figure EV3.

Referee #3 (R3):

In this report, De Guinoa et al show that mouse ILC3 express CD1d, present lipid antigens to invariant NKT cells (iNKT) and can be stimulated to produce IL-22 through CD1d engagement. The report is interesting but there are several flaws that diminish my enthusiasm for the study

1. The authors show that ILC3 can uptake lipid antigens *in vitro* and *in vivo*. It is unclear whether this is a real function of ILC3 or simply reflects a non-specific uptake of lipids. The authors show uptake by DC and ILC3. They should show whether other cells, like B cells and T cells, endocytose lipids to a similar extent in the same conditions or whether this is a unique feature of ILC3 and DC.

We apologize if this part of the paper has caused confusion. Lipid antigen uptake is not only restricted to DCs or ILC3s, and the ability of different cells to internalize lipid antigens has been broadly reported in the literature by us and others (Barral and Brenner, *Nat Rev Immunol* 2007; Salio *et al*, *Curr Opin Immunol* 2009; De Libero and Mori, *Trends in Immunol* 2012). As suggested, we have analyzed lipid uptake by B and T cells *in vitro* (see adjacent figure; red=control; black=T cells; blue=B cells) and confirmed that indeed they can acquire fluorescent lipids. We have clarified this and cited the relevant literature in our revised manuscript.

2. Which receptors mediate internalization of lipids in ILC3? No molecular basis for this phenomenon is provided

We agree with R3 that the understanding of the mechanisms mediating lipid internalization is an interesting question and we have discussed it in the revised manuscript.

Due to their hydrophobicity, lipids usually require the binding to proteins or lipoproteins for their transport in serum or extracellular environment, and this may determine the ability of different cells to internalise them. For instance, exogenous lipids can be incorporated into VLDL particles, which can then be internalized through LDL-Receptor mediated uptake and delivered to the endocytic system for CD1d presentation (van den Elzen *et al*, *Nature* 2005; Allan *et al*, *Blood* 2009). Other families of receptors such as scavenger receptors or the macrophage mannose receptors have also been implicated in the uptake of exogenous lipids (or lipid-containing complexes) and targeting towards the CD1d presentation pathway (Freigang *et al*, *J Clin Invest* 2012; Prigozy *et al*, *Immunity* 1997). Moreover, uptake of pathogens or particulate material by phagocytosis delivers exogenous antigens into the endocytic system, where CD1d molecules can bind them. Importantly, ILC3s can internalize proteins and 1 μ m latex beads (Hepworth *et al*, *Nature* 2013; von Burg *et al*, *PNAS* 2014) as well as lipids, which suggests that ILC3s may sample their environment and internalise antigens using a variety of mechanisms.

3. The authors propose that engagement of CD1d with antibody-mediated cross-linking induces IL-22. The intracellular increase of IL-22 is rather unimpressive. It is also a mystery how this can occur. IL-22 production is dependent on STAT signaling. Which signaling pathway is induced by CD1d? How do those signal activate the IL-22 promoter? Provided that anti-CD1d augments IL-22 secretion, it seems more likely that the antibody activates Fc receptors or acts through other mechanisms

We agree with R3 that the mechanism by which CD1d crosslink mediates cytokine production is an interesting question. Importantly, CD1d crosslink has been shown to induce cytokine secretion in several cell types including DCs, monocytes and intestinal epithelial cells (IEC, Olszak *et al*, *Nature* 2014; Colgan *et al*, *PNAS* 1999; Yue, *et al*, *PNAS* 2005; Yue *et al*, *J Immunol* 2010). Interestingly, CD1d engagement in IEC leads to STAT3 phosphorylation and STAT3-dependent cytokine secretion in a process that involves the intracellular tail of CD1d (Olszak *et al*, *Nature* 2014). It would be therefore possible for a similar STAT-dependent mechanism to regulate IL-22 production in response to CD1d crosslink on ILC3s. We have discussed this possibility in the revised manuscript.

4. Lipid loaded ILC3 activate iNKT *in vitro*. However, the authors don't provide any evidence that ILC3 are necessary for the activation of iNKT *in vivo*.

As suggested by R3 we have performed experiments that demonstrate that ILC3s can mediate lipid presentation and induce activation of iNKT cells *in vivo*. This new data is included in new Figure 2D-F.

5. *The authors show no RNA expression for CD1d. According to the available mRNA data from Immgen, CD1d RNA is almost undetectable in ILC3, suggesting that CD1d expression may be related to cross-dressing. The authors should address this possibility by directly measuring CD1d RNA amount*

We have measured CD1d mRNA in sorted ILC3s and included this new data in Figure 1C.

2nd Editorial Decision

15 July 2016

Thank you for your submission to EMBO reports. We have now received reports from the three referees that were asked to evaluate your study, which can be found at the end of this email. As you will see, two referees think that the study is now suitable for publication, whereas referee #3 still has concerns, in particular regarding the data shown in Fig. 3D.

After discussing with the referees we feel that for acceptance of the manuscript it would be important to show CD1d-mediated IL-22 production in a more convincing way. The referees suggest to measure IL-22 production after stimulation with CD1d in comparison with classical stimulation with IL-23, using intracellular staining and ELISA measurement of IL-22.

Given that all referees in principle agree on the potential interest of your study, we would like to give you the exceptional opportunity to revise your manuscript again, with the understanding that the remaining issues as highlighted above (and detailed in the review of referee #3) needs to be addressed in the final version of the manuscript.

Formally, papers in EMBO reports have to be accepted within 6 months of the initial decision, which in your case would be 21st October 2016 (otherwise the novelty would need to be re-assessed). Thus, it would be good that the revised manuscript is submitted within the next two months.

Regarding data quantification and statistics, can you please add statistics to all diagrams in the figures and specify the number "n" for how many experiments were performed, the bars and error bars (e.g. SEM, SD) and the test used to calculate p-values in the respective figure legends? This information must be provided in the figure legends. Please also include scale bars in all microscopy images.

We now strongly encourage the publication of original source data with the aim of making primary data more accessible and transparent to the reader. The source data will be published in a separate source data file online along with the accepted manuscript and will be linked to the relevant figure. If you would like to use this opportunity, please submit the source data (for example scans of entire gels or blots, data points of graphs in an excel sheet, additional images, etc.) of your key experiments together with the revised manuscript. Please include size markers for scans of entire gels, label the scans with figure and panel number, and send one PDF file per figure or per figure panel.

- a letter detailing your responses to the referee comments in Word format (.doc)
- a Microsoft Word file (.doc) of the revised manuscript text
- editable TIFF or EPS-formatted figure files in high resolution

In addition I would need from you:

- a short, two-sentence summary of the manuscript
- two to three bullet points highlighting the key findings of your study
- a schematic summary figure (in jpeg or tiff format with the exact width of 550 pixels and a height of about 400 pixels) that can be used as visual synopsis on our website.

Please note that we now mandate that all corresponding authors list an ORCID digital identifier!

I look forward to seeing a revised version of your manuscript when it is ready. Please let me know if

you have questions or comments regarding the revision.

REFEREE REPORTS

Referee #1:

The revised manuscript presented by Suarez de Guinoa is acceptable. The authors replied carefully to all comments and concerns and made appropriate changes in the manuscript and the figures. The manuscript is therefore clearer and goes more in depth, which strengthen the conclusions. As a reviewer, I think that this manuscript is now suitable for publication in EMBO reports.

Referee #2:

The authors adequately addressed my comments, except my suggestion to confirm CD1d expression in human ILC3s. Lack of human data should be acknowledged in the Discussion, which may further benefit from a scrutiny of available human ILC microarray data. Overall, this is an elegant, clear and well-executed work that deserves publication.

Referee #3:

The revised paper is not substantially improved. The entire message of the paper (CD1d triggers IL-22 production by ILC3), relies on Fig. 3D. As pointed out in my original review, this plot is totally unconvincing and unacceptable; it does not show reliable production of IL-22. What I consider reliable, for comparison, is the intracellular staining shown in the plot presented in Fig. 2F. The uptake of lipids seems nonspecific, as it occurs in any cell tested. In addition, no mechanism is implicated for such uptake - the authors discuss scavenger receptors, phagocytosis and LDL receptors but show no data for any of them. The authors suggest that CD1d may trigger release of IL-22 through STAT3 but provide no evidence for this. Finally, while I appreciate the attempt to show that ILC3s induce NKT cell activation *in vivo*, the new experiment presented shows that ILC3s CAN induce NKT cells in a contrived system, but it does not show that ILC3s are NECESSARY for NKT cell activation.

2nd Revision - authors' response

07 September 2016

We have performed additional stimulation experiments with ILC3s and now show that CD1d engagement induces IL-22 as detected by qPCR, flow-cytometry and ELISA. Moreover we demonstrate that CD1d stimulation acts synergistically with IL-23 to induce increased IL-22 production. This new data has been included in Figure 3D-G

Referee #1:

The revised manuscript presented by Suarez de Guinoa is acceptable. The authors replied carefully to all comments and concerns and made appropriate changes in the manuscript and the figures. The manuscript is therefore clearer and goes more in depth, which strengthen the conclusions. As a reviewer, I think that this manuscript is now suitable for publication in EMBO reports.

We thank the reviewer for his/her comments

Referee #2:

The authors adequately addressed my comments, except my suggestion to confirm CD1d expression in human ILC3s. Lack of human data should be acknowledged in the Discussion, which may further benefit from a scrutiny of available human ILC microarray data. Overall, this is an elegant, clear and well-executed work that deserves publication.

We thank the reviewer for his/her comments. We agree that analyzing the expression of the CD1 isoforms in human ILCs is an important question, and we have acknowledged this in the discussion.

Referee #3:

The revised paper is not substantially improved. The entire message of the paper (CD1d triggers IL-22 production by ILC3), relies on Fig. 3D. As pointed out in my original review, this plot is totally unconvincing and unacceptable; it does not show reliable production of IL-22. What I consider reliable, for comparison, is the intracellular staining shown in the plot presented in Fig. 2F.

The uptake of lipids seems nonspecific, as it occurs in any cell tested. In addition, no mechanism is implicated for such uptake - the authors discuss scavenger receptors, phagocytosis and LDL receptors but show no data for any of them.

The authors suggest that CD1d may trigger release of IL-22 through STAT3 but provide no evidence for this.

Finally, while I appreciate the attempt to show that ILC3s induce NKT cell activation in vivo, the new experiment presented shows that ILC3s CAN induce NKT cells in a contrived system, but it does not show that ILC3s are NECESSARY for NKT cell activation.

We have performed additional stimulation experiments with ILC3s and now we show that CD1d engagement induces IL-22 as detected by qPCR, flow-cytometry and ELISA (Figure 3D-G). We agree with Reviewer 3 that our IL-22 intracellular staining is not as strong as that observed with IFN γ produced by iNKT cells, which is probably due to lower cytokine production by ILC3s in comparison with α GalCer-stimulated iNKT cells, as well as lower affinity of commercially available α IL-22 antibodies. However although smaller, our IL-22 staining is reproducible. We have repeated our flow-cytometry experiments 8 independent times getting similar results. Moreover our flow-cytometry data correlates with the results obtained by qPCR and ELISA, for stimulations with α CD1d, IL-23 and the combination of both (Figure 3D-G).

In the current manuscript we provide proof-of-concept evidence for CD1d expression, lipid internalization and presentation capacity by ILC3s *in vitro* and *in vivo*. Moreover we show that CD1d engagement leads to ILC3 activation and IL-22 production. Although we agree with R3 that addressing the mechanisms underlying these processes are important questions, getting conclusive answers to these questions would require a considerable amount of time and resources (i.e. transgenic/KO mice) resulting in a very significant delay for publication of our findings. We believe that mechanistic studies related to ILC3s should be the focus of future manuscripts.

We agree with the reviewer that our *in vivo* experiments don't demonstrate that ILC3s are necessary for iNKT cell activation, but we have never intended to say that they are. As we pointed out in the previous letter, our data demonstrates that ILC3s have the capacity to induce activation of iNKT cells *in vivo*. Further studies using mice lacking CD1d exclusively on ILC3s would provide the most accurate approach to specifically define the role of ILC3s in iNKT cell activation, establishing if they are indispensable for iNKT cell activation and/or if they act synergistically with other APCs to modulate iNKT cell functions.

Accepted

04 October 2016

I am very pleased to accept your manuscript for publication in the next available issue of EMBO reports. Thank you for your contribution to our journal.

At the end of this email I include important information about how to proceed. Please ensure that you take the time to read the information and complete and return the necessary forms to allow us to publish your manuscript as quickly as possible.

As part of the EMBO publication's Transparent Editorial Process, EMBO reports publishes online a Review Process File to accompany accepted manuscripts. As you are aware, this File will be published in conjunction with your paper and will include the referee reports, your point-by-point response and all pertinent correspondence relating to the manuscript.

If you do NOT want this File to be published, please inform the editorial office within 2 days, if you have not done so already, otherwise the File will be published by default [contact: emboreports@embo.org]. If you do opt out, the Review Process File link will point to the following statement: "No Review Process File is available with this article, as the authors have chosen not to

make the review process public in this case."

Thank you again for your contribution to EMBO reports and congratulations on a successful publication. Please consider us again in the future for your most exciting work.

REFEREE REPORT

Referee #3

The authors have addressed my concerns to the best of their abilities. Although I remain skeptical that CD1 can induce even more IL-22 than IL-23 stimulation, the paper is novel and provocative and will stimulate discussions and further studies in the field.

Corresponding Author Name: Patricia Barral

Manuscript Number: EMBOR-2016-42412